# Representing Mixtures of Word Embeddings with Mixtures of Topic Embeddings

**Dongsheng Wang**[1],[*] **Dandan Guo**[2],[*] **He Zhao**[3], **Huangjie Zheng**[4],
**Korawat Tanwisuth**[4], **Bo Chen**[1], **Mingyuan Zhou**[4]
[1]Xidian University    [2]The Chinese University of Hong Kong, Shenzhen
[3]Monash University    [4]The University of Texas at Austin
`wds@stu.xidian.edu.cn, guodandan@cuhk.edu.cn`
`ethan.zhao@monash.edu, huangjie.zheng@utexas.edu`
`korawat.tanwisuth@utexas.edu, bchen@mail.xidian.edu.cn`
`mingyuan.zhou@mccombs.utexas.edu`

## Abstract

A topic model is often formulated as a generative model that explains how each word of a document is generated given a set of topics and document-specific topic proportions. It is focused on capturing the word co-occurrences in a document and hence often suffers from poor performance in analyzing short documents. In addition, its parameter estimation often relies on approximate posterior inference that is either not scalable or suffering from large approximation error. This paper introduces a new topic-modeling framework where each document is viewed as a set of word embedding vectors and each topic is modeled as an embedding vector in the same embedding space. Embedding the words and topics in the same vector space, we define a method to measure the semantic difference between the embedding vectors of the words of a document and these of the topics, and optimize the topic embeddings to minimize the expected difference over all documents. Experiments on text analysis demonstrate that the proposed method, which is amenable to mini-batch stochastic gradient descent based optimization and hence scalable to big corpora, provides competitive performance in discovering more coherent and diverse topics and extracting better document representations.

## 1  Introduction

For text analysis, topic models are widely used to extract a set of latent topics from a corpus (a collection of documents). The extracted topics, revealing common word co-occurrence patterns within a document, often correspond to semantically meaningful concepts in the training corpus. Bayesian probabilistic topic models (BPTMs), such as latent Dirichlet allocation (LDA) (Blei et al., 2003; Griffiths & Steyvers, 2004) and its nonparametric Bayesian generalizations (Teh et al., 2006; Zhou et al., 2012), have been the most popular ones. A BPTM is often formulated as a generative model that explains how each word of a document is generated given a set of topics and document-specific topic proportions. Bayesian inference of a BPTM is usually based on Gibbs sampling or variational inference (VI), which can be less scalable for big corpora and need to be customized accordingly.

With the recent development in auto-encoding VI, originated from variational autoencoders (VAEs) (Kingma & Welling, 2014; Rezende et al., 2014), deep neural networks have been successfully used to develop neural topic models (NTMs) (Miao et al., 2016; Srivastava & Sutton, 2017; Burkhardt & Kramer, 2019; Zhang et al., 2018; Dieng et al., 2020; Zhao et al., 2021). The key advantage of NTMs is that approximate posterior inference can be carried out easily via a forward pass of the encoder network, without the need for expensive iterative inference scheme per test observation as in both Gibbs sampling and conventional VI. Hence, NTMs enjoy better flexibility and scalability than BPTMs. However, the reparameterization trick in VAEs cannot be directly applied to the Dirichlet (Burkhardt & Kramer, 2019) or gamma distributions (Zhang et al., 2018), which are usually used as

---

[*]Equal contribution.

the prior and conditional posterior of latent topics and topic proportions, so approximations have to be used, potentially introducing additional complexity or approximation errors.

To address the above shortcomings, we propose a novel topic modeling framework in an intuitive and effective manner of enjoying several appealing properties over previously developed BPTMs and NTMs. Like other TMs, we also focus on learning the global topics shared across the corpus and the document-specific topic proportions, which are the two key outputs of a topic model. Without building an explicit generative process, we formulate the learning of a topic model ($e.g.$, optimizing the likelihood) as the process of minimizing the distance between each observed document $j$ and its corresponding trainable distribution. More specifically, the former (document $j$) can be regarded as as an empirical discrete distribution $P_j$, which has an uniform measure over all the words within this document. To construct the latter (trainable distribution), we can represent $P_j$ with $K$ shared topics and its $K$-dimensional document-specific topic proportion, defined as $Q_j$, where we view shared topics as $K$ elements and topic proportion as the probability measure in $Q_j$. It is very reasonable since the $k$-th element in topic proportion measures the weight of topic $k$ for a document, and the document can be represented perfectly using the learned topic proportion and topics from a desired TM. Recalling that each topic and word usually reside in the $V$-dimensional (vocabulary size) space in TMs, it might be difficult to directly optimize the distance between $P_j$ and $Q_j$ over $V$-dimensional space. Motivated by Dieng et al. (2020), we further assume that both topics and words live in a $H$-dimensional embedding space, much smaller than the vocabulary space. With a slight abuse of notation, we still use $P_j$ over the word embeddings and $Q_j$ over the topic embeddings as two representations for document $j$. Below, we turn towards pushing the document-specific to-be-learned distribution $Q_j$ to be as close as possible to the empirical distribution $P_j$.

To this end, we develop a probabilistic bidirectional transport-based method to measure the semantic difference between the two discrete distributions in an embedding space. By minimizing the expected difference between $P_j$ and $Q_j$ over all documents, we can learn the topic and word embeddings directly. Importantly, we naturally leverage semantic distances between topics and words in an embedding space to construct the point-to-point cost of moving between them, where the cost becomes a function of topic embeddings. Notably, we consider linking the word embeddings in $P_j$ and topic embeddings in $Q_j$ in a bidirectional view. That is, given a word embedding drawn from $P_j$, it is more likely to be linked to a topic embedding that both is closer to it in the embedding space and exhibits a larger proportion in $Q_j$; vice versa. Our proposed framework has several key properties: **1)** By bypassing the generative process, our proposed framework avoids the burden of developing complex sampling schemes or approximations for the posterior of BPTMs or NTMs. **2)** The design of our proposed model complies with the principles of TMs, whose each learned topic describes an interpretable semantic concept. More interestingly, our model is flexible to learn word embeddings from scratch or use/finetune pretrained word embeddings. When pretrained word embeddings are used, our model naturally alleviates the issue of insufficient word co-occurrence information in short texts as discussed by prior work (Dieng et al., 2020; Zhao et al., 2017a; 2021), which is one of the key drawbacks of many BPTMs and NTMs. **3)** Conventional TMs usually enforce a simplex constraint on the topics over a fixed vocabulary, which hinders their applications in the case where the vocabulary varies. In our method, we view a document as a mixture of word embedding vectors, which facilitates the deployment of the model when the size of the vocabulary varies. Finally, we have conducted comprehensive experiments on a wide variety of datasets in the comparison with advanced BPTMs and NTMs, which show that our model can achieve the state-of-the-art performance as well as appleaing interpretability. The code is available at https://github.com/BoChenGroup/WeTe.

## 2 BACKGROUND

**Topic Models:** TMs usually represent each document in a corpus as a bag-of-words (BoW) count vector $\boldsymbol{x} \in \mathbb{R}_+^V$, where $x_v$ represents the occurrences of word $v$ in the vocabulary of size $V$. A TM aims to discover $K$ topics in the corpus, each of which describes a specific semantic concept. A topic is or can be normalized into a distribution over the words in the vocabulary, named word distribution, $\boldsymbol{\phi}_k \in \Sigma_V$, where $\Sigma_V$ is a $V-1$ dimensional simplex and $\boldsymbol{\phi}_{vk}$ indicates the weight or relevance of word $v$ under topic $k$. Each document comes from a mixture of topics, associated with a specific mixture proportion, which can be captured by a distribution over $K$ topics, named topic proportion, $\boldsymbol{\theta} \in \Sigma_K$, where $\theta_k$ indicates the weight of topic $k$ for a document.

As the most fundamental and popular series of TMs, BPTMs (Blei et al., 2003; Zhou et al., 2012; Hoffman et al., 2010) generate the document $x$ with latent variables (*i.e.*, topics $\{\phi_k\}_{k=1}^K$ and topic proportion $\theta$) sampled from pre-specified prior distributions (*e.g.*, gamma and Dirichlet). Like other Bayesian models, the learning process of a BPTM relies on Bayesian inference, such as variational inference and Gibbs sampling. Recently, NTMs (Zhang et al., 2018; Burkhardt & Kramer, 2019; Dieng et al., 2020; Wang et al., 2020; Duan et al., 2021b) have attracted significant research interests in topic modeling. Most existing NTMs can be regarded as extensions of BPTM like LDA within the VAEs framework (Zhao et al., 2021). In general, NTMs consist of an encoder network that maps the (normalized) BoW input $x$ to its topic proportion $\theta$, and a decoder network that generates $x$ conditioned on the topics $\{\phi_k\}_{k=1}^K$ and proportion $\theta$. Despite their appealing flexibility and scalability, due to the unusable reparameterization trick in original VAEs for the Dirichlet or gamma distributions, NTMs have to develop complex sampling schemes or approximations, leading to potentially large approximation errors or learning complexity.

**Compare Two Discrete Distributions:** This paper aims to quantify the difference between two discrete distributions (word embeddings and topic embeddings), whose supports are points in the same embedding space. Specifically, let $p$ and $q$ be two discrete probability measures on the arbitrary space $X \subseteq \mathbb{R}^H$, formulated as $p = \sum_{i=1}^n u_i \delta_{x_i}$ and $q = \sum_{j=1}^m v_j \delta_{y_j}$, where $\boldsymbol{u} = [u_i] \in \Sigma_n$ and $\boldsymbol{v} = [v_j] \in \Sigma_m$ denote two distributions of the discrete states. To measure the distance between $p$ and $q$, a non-trivial way is to use optimal transport (OT) (Peyré & Cuturi, 2019) defined as

$$\text{OT}(p, q) := \min_{\mathbf{T} \in \Pi(\boldsymbol{u}, \boldsymbol{v})} \text{Tr}\left(\mathbf{T}^\top \boldsymbol{C}\right), \tag{1}$$

where $\boldsymbol{C} \in \mathbb{R}_{\geq 0}^{n \times m}$ is the transport cost matrix with $C_{ij} = c(x_i, y_j)$, $\mathbf{T} \in \mathbb{R}_{>0}^{n \times m}$ a doubly stochastic transport matrix such that $\Pi(\boldsymbol{u}, \boldsymbol{v}) = \left\{\mathbf{T} \mid \mathbf{T}\mathbf{1}_{D_v} = \boldsymbol{u}, \mathbf{T}^\top \mathbf{1}_{D_u} = \boldsymbol{v}\right\}$, $T_{ij}$ the transport probability between $x_i$ and $y_j$, and $\text{Tr}(\cdot)$ the matrix trace. Since the transport plan is imposed on the constraint of $\mathbf{T} \in \Pi(\boldsymbol{u}, \boldsymbol{v})$, it has to be computed via constrained optimizations, such as the iterative Sinkhorn algorithm when an additional entropy regularization term is added (Cuturi, 2013).

The recently introduced conditional transport (CT) framework (Zheng & Zhou, 2020; Tanwisuth et al., 2021) can be used to quantify the difference between two discrete distributions, which, like OT, does not require the distributions to share the same support. CT considers the transport plan in a bidirectional view, which consists of a forward transport plan as $\mathbf{T}^{p \to q}$ and backward transport plan $\mathbf{T}^{p \leftarrow q}$. Therefore, the transport cost between two empirical distributions in CT can be expressed as

$$\text{CT}(p, q) := \min_{\mathbf{T}^{p \to q}, \mathbf{T}^{q \to p}} \text{Tr}\left[(\mathbf{T}^{p \to q})^\top \boldsymbol{C} + (\mathbf{T}^{q \to p})^\top \boldsymbol{C}\right]. \tag{2}$$

CT specifies $\mathbf{T}_{ij}^{p \to q} = u_i \frac{v_j e^{-d_\psi(y_j, x_i)}}{\sum_{j'=1}^m v_{j'} e^{-d_\psi(x_i, y_{j'})}}$ and $\mathbf{T}_{ij}^{p \leftarrow q} = v_j \frac{u_i e^{-d_\psi(y_j x_i)}}{\sum_{i'=1}^n u_{i'} e^{-d_\psi(x_{i'}, y_j)}}$ and hence $\mathbf{T}^{p \to q} \mathbf{1}_{D_v} = \boldsymbol{u}$ and $(\mathbf{T}^{p \leftarrow q})^T \mathbf{1}_{D_u} = \boldsymbol{v}$, but in general $\mathbf{T}^{p \leftarrow q} \mathbf{1}_{D_v} \neq \boldsymbol{u}$ and $(\mathbf{T}^{p \to q})^T \mathbf{1}_{D_u} \neq \boldsymbol{v}$. Here $d_\psi(x, y) = d_\psi(y, x)$ parameterized by $\psi$ measures the difference between two vectors. This flexibility of CT potentially facilitates an easier integration with deep neural networks with a lower complexity and better scalability. These properties can be helpful to us in the development of a new topic modeling framework based on transportation between distributions, especially for NTMs.

## 3 LEARNING MIXTURE OF TOPIC EMBEDDINGS

Now we will describe the details of the proposed model. Since it represents a mixture of **W**ord **E**mbeddings as a mixture of **T**opic **E**mbeddings, we refer to it as WeTe. Specifically, consider a corpus of $J$ documents, where the vocabulary contains $V$ distinct terms. Unlike in other TMs, where a document is represented as a BoW count vector $x \in \mathbb{R}_+^V$ as shown in Section 2, we denote each document as a set of its words, defined as $D_j = [w_{ji}]$, where $w_{ji} \in \{1, \ldots, V\}$ means the $i$-th word in the $j$-th document with $i \in [1, N_j]$, and $N_j$ is the length of the $j$-th document. Denote $\mathbf{E} \in \mathbb{R}^{H \times V}$ as the word embedding matrix whose columns contain the embedding representations of the terms in the vocabulary. By projecting each word into the corresponding word-embedding space, we thus represent each document as an empirical distribution $P_j$ on the word embedding space as follows

$$P_j = \sum_{i=1}^{N_j} \frac{1}{N_j} \delta_{\boldsymbol{w}_{ji}}, \ \boldsymbol{w}_{ji} \in \mathbb{R}^H. \tag{3}$$

Similar to other TMs, we aim to learn $K$ topics from the corpus. However, instead of representing a topic as a distribution over the terms in the vocabulary, we use an embedding vector for each topic,

$\boldsymbol{\alpha}_k \in \mathbb{R}^H$. Here topic embedding $\boldsymbol{\alpha}_k$ is a distributed representation of the $k$-th topic in the same semantic space of the word embeddings. Collectively, we form a document-specific empirical topic distribution $Q_j$ (on the embedding space), defined as

$$Q_j = \sum_{k=1}^K \tilde{\theta}_{jk} \delta_{\boldsymbol{\alpha}_k}, \quad \boldsymbol{\alpha}_k \in \mathbb{R}^H. \tag{4}$$

Here $\tilde{\theta}_{j,1:K}$ denotes the normalized topic proportions of document $j$, i.e., $\tilde{\boldsymbol{\theta}}_j := \boldsymbol{\theta}_j / \sum_{k=1}^K \theta_{jk}$. We focus on learning the topic distribution $Q_j$ that is close to distribution $P_j$. Exploiting the CT loss defined in Eq. (2), we introduce WeTe as a novel topic model for text analysis. For document $j$, we propose to minimize the expected difference between the word embeddings from $P_j$ and topic embeddings from $Q_j$ in terms of its topic proportion and topic embeddings. For all the documents in the corpus, we can minimize the average CT loss

$$\min_{\boldsymbol{\alpha},\boldsymbol{\Theta}} \frac{1}{J} \sum_{j=1}^J [\mathrm{CT}(P_j, Q_j)]. \tag{5}$$

As a bidirectional transport, $\mathrm{CT}(\cdot)$ consists of a doc-to-topic CT that transports the word embeddings to topic embeddings, and a topic-to-doc CT that reverses the transport direction. We define a conditional distribution specifying how likely a given topic embedding $\boldsymbol{\phi}_k$ will be transported to word embedding $\boldsymbol{\alpha}_{ji}$ in document $j$ as

$$\pi_{N_j}(\boldsymbol{w}_{ji} \,|\, \boldsymbol{\alpha}_k) = \frac{P_j(\boldsymbol{w}_{ji}) e^{-d(\boldsymbol{w}_{ji},\boldsymbol{\alpha}_k)}}{\sum_{i'=1}^{N_j} P_j(\boldsymbol{w}_{ji'}) e^{-d(\boldsymbol{w}_{ji'},\boldsymbol{\alpha}_k)}} = \frac{e^{-d(\boldsymbol{w}_{ji},\boldsymbol{\alpha}_k)}}{\sum_{i'=1}^{N_j} e^{-d(\boldsymbol{w}_{ji'},\boldsymbol{\alpha}_k)}}, \quad \boldsymbol{w}_{ji} \in \{\boldsymbol{w}_{j1}, \ldots, \boldsymbol{w}_{jN_j}\}, \tag{6}$$

where $d(\boldsymbol{w}_i, \boldsymbol{\alpha}_k) = d(\boldsymbol{\alpha}_k, \boldsymbol{w}_i)$ indicates the semantic distance between the two vectors. Intuitively, if $\boldsymbol{\alpha}_k$ and $\boldsymbol{w}_{ji}$ have a small semantic distance, the $\pi(\boldsymbol{w}_i \,|\, \boldsymbol{\alpha}_k)$ would have a high probability. This construction makes it easier to transport $\boldsymbol{\alpha}_k$ to a word that is closer to it in the embedding space. For document $j$ with $N_j$ words $\{\boldsymbol{w}_{j1}, \ldots, \boldsymbol{w}_{jN_j}\}$, the topic-to-doc CT cost can be expressed as

$$L_{Q_j \to P_j} = \mathbb{E}_{\boldsymbol{\alpha}_k \sim Q_j} \mathbb{E}_{\boldsymbol{w}_{ji} \sim \pi_{N_j}(\cdot \,|\, \boldsymbol{\alpha}_k)}[c(\boldsymbol{w}_{ji}, \boldsymbol{\alpha}_k)] = \sum_{k=1}^K \tilde{\theta}_{jk} \sum_{i=1}^{N_j} c(\boldsymbol{w}_{ji}, \boldsymbol{\alpha}_k) \pi(\boldsymbol{w}_{ji} \,|\, \boldsymbol{\alpha}_k), \tag{7}$$

where $c(\boldsymbol{w}_{ji}, \boldsymbol{\alpha}_k) = c(\boldsymbol{\alpha}_k, \boldsymbol{w}_{ji}) \geq 0$ denotes the point-to-point cost of transporting between word embedding $\boldsymbol{w}_{ji}$ and topic embedding $\boldsymbol{\alpha}_k$, and $\tilde{\theta}_{k,j}$ can be considered as the weight of transport cost between all words in document $j$ and topic embedding $k$. Similar to but different from Eq. (7), we introduce the doc-to-topic CT, whose transport cost is defined as

$$L_{P_j \to Q_j} = \mathbb{E}_{\boldsymbol{w}_{ji} \sim P_j} \mathbb{E}_{\boldsymbol{\alpha}_k \sim \pi_K(\cdot \,|\, \boldsymbol{w}_{ji})}[c(\boldsymbol{w}_{ji}, \boldsymbol{\alpha}_k)] = \sum_{i=1}^{N_j} \frac{1}{N_j} \sum_{k=1}^K c(\boldsymbol{w}_{ji}, \boldsymbol{\alpha}_k) \pi(\boldsymbol{\alpha}_k \,|\, \boldsymbol{w}_{ji}), \tag{8}$$

where $\frac{1}{N_j}$ denotes the weight of transport cost between word $\boldsymbol{w}_{ji}$ in document $j$ and topic embeddings. In contrast to Eq. (6), we define the conditional transport probability from word embedding $\boldsymbol{w}_{ji}$ in document $j$ to a topic embedding $\boldsymbol{\phi}_k$ with

$$\pi(\boldsymbol{\alpha}_k \,|\, \boldsymbol{w}_{ji}) = \frac{Q_j(\boldsymbol{\alpha}_k) e^{-d(\boldsymbol{w}_{ji},\boldsymbol{\alpha}_k)}}{\sum_{k'=1}^K Q_j(\boldsymbol{\alpha}_{k'}) e^{-d(\boldsymbol{w}_{ji'},\boldsymbol{\alpha}_{k'})}} = \frac{e^{-d(\boldsymbol{w}_{ji},\boldsymbol{\alpha}_k)} \tilde{\theta}_{jk}}{\sum_{k'=1}^K e^{-d(\boldsymbol{w}_{ji},\boldsymbol{\alpha}_{k'})} \tilde{\theta}_{jk'}}, \tag{9}$$

where $\tilde{\theta}_{jk} = Q_j(\boldsymbol{\alpha}_k)$ can be interpreted as the prior weight of topic embedding $\boldsymbol{\alpha}_k$ in document $j$.

We have not specified the form of $c(\boldsymbol{w}_{ji}, \boldsymbol{\alpha}_k)$ and $d(\boldsymbol{w}_{ji}, \boldsymbol{\alpha}_k)$. A naive definition of the transport cost or semantic distance between two points is some distance between their raw feature vectors. In our framework, we specify the following construction of cost function:

$$c(\boldsymbol{w}_{ji}, \boldsymbol{\alpha}_k) = e^{-\boldsymbol{w}_{ji}^{\mathrm{T}} \boldsymbol{\alpha}_k}. \tag{10}$$

Here the cost function is defined for two reasons: the inner product is the commonly-used way to measure the difference between two embedding vectors, and the cost needs to be positive. For semantic distance, we directly take the inner product of the word embedding $\boldsymbol{w}_{ji}$ and the topic embedding $\boldsymbol{\alpha}_k$, i.e., $d(\boldsymbol{w}_{ji}, \boldsymbol{\alpha}_k) = -\boldsymbol{w}_{ji}^{\mathrm{T}} \boldsymbol{\alpha}_k$, although other choices are possible.

### 3.1 REVISITING OUR PROPOSED MODEL FROM TOPIC MODELS

Generally speaking, conditioned on an observed document, traditional TMs (Blei et al., 2003; Zhou et al., 2012; Srivastava & Sutton, 2017) often decompose the distribution of the document's words into two learnable factors: the distribution of words conditioned on a certain topic, and the distribution

of topics conditioned on the document. Here, we establish the connection between our model and traditional TMs. Recall that $d(\boldsymbol{w}_i, \boldsymbol{\alpha}_k)$ indicates the semantic distance between topic $k$ and word $i$ in the embedding space. For arbitrary $\boldsymbol{\alpha}_k$ and $\boldsymbol{w}_i$, the more similar they are, the smaller underlying distance they have. Following this viewpoint, we assume $\boldsymbol{\phi}_k \in \mathbb{R}_+^V$ as the distribution-over-words representation of topic $k$ and treat its elements as

$$\boldsymbol{\phi}_{vk} := \frac{e^{-d(\boldsymbol{v}_v, \boldsymbol{\alpha}_k)}}{\sum_{v'=1}^V e^{-d(\boldsymbol{v}_{v'}, \boldsymbol{\alpha}_k)}}, \tag{11}$$

where $\boldsymbol{v}_v \in \mathbb{R}^H$ denotes the embedding of the $v$-th term in the vocabulary. Therefore, the column vector $\boldsymbol{\phi}_k$ weights the word importance in the corresponding topic $k$. With this form, our proposed model assigns a probability to a word in topic $k$ by measuring the agreement between the word's and topic's embeddings. Conditioned on $\boldsymbol{\Phi} = [\boldsymbol{\phi}_k]$, the flexibility of CT enables multiple ways to learn or define the topic proportions of documents, $i.e.$, $\boldsymbol{\Theta}$ detailed in Section 3.2. With CT's ability for modeling geometric structures, our model avoids developing the prior/posterior distributions and the associated sampling schemes, which are usually nontrivial in traditional TMs (Zhou et al., 2016).

## 3.2 Learning topic embeddings and topic proportions

Given the corpus of $J$ documents, we wish to learn the topic embedding matrix $\boldsymbol{\alpha}$ and topic proportions of documents $\boldsymbol{\Theta}$. Based on the doc-to-topic and topic-to-doc CT losses and the definitions of $c$ and $d$ in Eq. (6-10) and $\sum_{k=1}^K \tilde{\theta}_{jk} = 1$, we can rewrite the CT loss in Eq. 5 as

$$\frac{1}{J} \sum_{j=1}^J \mathrm{CT}(P_j, Q_j) = \frac{1}{J} \sum_{j=1}^J \left[ \left( \sum_{k=1}^K \frac{\tilde{\theta}_{jk}}{\sum_{i'=1}^{N_j} e^{w_{ji'}^T \boldsymbol{\alpha}_k \frac{1}{N_j}}} \right) + \left( \sum_{i=1}^{N_j} \frac{\frac{1}{N_j}}{\sum_{k'=1}^K e^{w_{ji}^T \boldsymbol{\alpha}_{k'} \tilde{\theta}_{jk'}}} \right) \right], \tag{12}$$

whose detailed derivation is shown in Appendix A. The two terms in the bracket exhibit appealing symmetry properties between the normalized topic proportion $\tilde{\boldsymbol{\theta}}_j$ and word prior $\frac{1}{N_j}$. To minimize the first term, for a given document whose topic proportion has a non-negligible activation at the $k$-th topic, the inferred $k$-th topic needs to be close to at least one word (in the embedding space) of that document. Similarly each word in document $j$ needs to find at least a single non-negligibly-weighted topic that is sufficiently close to it. In other words, the learned topics are expected to have a good coverage of the word embedding space occupied by the corpus by optimizing those two terms.

Like other TMs, the latent representation of the document is a distribution over $K$ topics: $\tilde{\boldsymbol{\theta}}_j \in \Sigma^K$, each element of which denotes the proportion of one topic in this document. Previous work shows that the data likelihood can be helpful to regularize the optimization of the a transport based loss (Frogner et al., 2015; Zhao et al., 2021). To amortize the computation of $\boldsymbol{\theta}_j$ and provide additional regularization, we introduce a regularized CT loss as

$$\min_{\boldsymbol{\alpha}, \mathbf{W}} \frac{1}{J} \sum_{j=1}^J \mathbb{E}_{\boldsymbol{\theta}_j \sim q_{\boldsymbol{W}}(\cdot \,|\, \boldsymbol{x}_j)} \left[ \mathrm{CT}(P_j, Q_j) - \epsilon \log p(\boldsymbol{x}_j; \boldsymbol{\alpha}, \boldsymbol{\theta}_j) \right], \tag{13}$$

where $q_{\boldsymbol{W}}(\boldsymbol{\theta}_j \,|\, \boldsymbol{x}_j)$ is a deterministic or stochastic encoder, parameterized by $\boldsymbol{W}$, $p(\boldsymbol{x}_j; \boldsymbol{\Phi}, \boldsymbol{\theta}_j) = \mathrm{Poisson}(\boldsymbol{x}_j; \sum_{k=1}^K \boldsymbol{\phi}_k \theta_{jk})$ ($\boldsymbol{\phi}_k$ is defined as in Eq. (11)) is the likelihood used in Poisson factor analysis (Zhou et al., 2012), and $\epsilon$ is a trade-off hyperparameter between the CT loss and log-likelihood. Here, we encode $\boldsymbol{\theta}$ with the Weibull distribution: $q_{\boldsymbol{W}}(\boldsymbol{\theta}_j \,|\, \boldsymbol{x}_j) = \mathrm{Weibull}(f_{\boldsymbol{W}}(\boldsymbol{x}_j), g_{\boldsymbol{W}}(\boldsymbol{x}_j))$, where $f$ and $g$ are two related neural networks parameterized by $\boldsymbol{W}$. Similar to previous work (Zhang et al., 2018; Guo et al., 2020; Duan et al., 2021a), we choose Weibull mainly because it resembles the gamma distribution and is reparameterizable, as drawing $m \sim \mathrm{Weibull}(k, \lambda)$ is equivalent to mapping $m = \hat{f}(\epsilon) := \lambda(-log(1-\epsilon))^{1/k}$, $\epsilon \sim \mathrm{Uniform}(0,1)$. Different from previous work, here $q_{\boldsymbol{W}}(\boldsymbol{\theta}_j \,|\, \boldsymbol{x}_j)$ does not play the role of a variational inference network that aims to approximate the posterior distribution given the likelihood and a prior. Instead, it is encouraged to strike a balance between minimizing the CT cost, between the document representation in the word embedding space and that in the topic embedding space, and minimizing the negative log-likelihood of Poisson factor analysis, with the document representation shared between both components of the loss.

The loss of Eq. (13) is differentiable in terms of $\boldsymbol{\alpha}$ and $\mathbf{W}$, which can be optimized jointly in one training iteration. The training algorithm is outlined in Appendix B. Benefiting from the encoding network, after training the model, we can obtain $\boldsymbol{\theta}_j$ by mapping the new input $\boldsymbol{x}_j$ with the learned encoder $\mathbf{W}$, avoiding the hundreds iterations in MCMC or VI to collect posterior samples for

local variables. The algorithm for WeTe can either use pretrained word embeddings, $e.g.$, GloVe (Pennington et al., 2014), or learn them from scratch. Practically speaking, using pretrained word embeddings enables more efficient learning for reducing the parameter space, and has been proved beneficial for short documents for leveraging the rich semantic information in pretrained word embeddings. In our experiments, WeTe by default uses the GloVe word embeddings.

# 4 RELATED WORK

**Models with Word Embeddings:** Word embeddings have been widely used as complementary information to improve topic models. Skipgram-based models (Shi et al., 2017; Moody, 2016; Park & Lee, 2020) jointly model the skip-gram word embeddings and latent topic distributions under the Skipgram Negative-Sampling objective. Those models incorporate the topical context into the central words to generate its surrounding words, which share similar idea with the topic-to-doc transport in WeTe that views the topic vectors as the central words, and words within a document as the surrounding words. Besides, WeTe forces the inferred topics to be close to at least one word embedding vector of a given document by the doc-to-topic transport, which is not considered in those models. For BPTMs, word embeddings are usually incorporated into the generative process of word counts (Petterson et al., 2010; Nguyen et al., 2015; Li et al., 2016; Zhao et al., 2017a;b; Keya et al., 2019). Benefiting from the flexibility of NTMs, word embeddings can be either incorporated as part of the encoder input, such as in Card et al. (2018), or used in the generative process of words, such as in Dieng et al. (2020). Because these models construct the explicit generative processes from the latent topics to documents and belong to the extensions of BPTMs or NTMs, they may still face these previously mentioned difficulties in TMs. Our method naturally incorporates word embeddings into the distances between topics and words under the bidirectional transport framework, which is different from previous ones.

**Models by Minimizing Distances of Distributions:** Yurochkin et al. (2019) adopt OT to compare two documents' similarity between their topic distributions extracted from a pretrained LDA, but their focus is not to learn a topic model. Nan et al. (2019) extend the framework of Wasserstein autoEncoders (Tolstikhin et al., 2018) to minimize the Wasserstein distance between the fake data generated with topics and real data, which can be interpreted as an OT variant to NTMs based on VAE. In addition, Xu et al. (2018) introduce distilled Wasserstein learning, where an observed document is approximated with the weighted Wasserstein barycentres of all the topic-word distributions and the weights are viewed as the topic proportion of that document. The Optimal Transport based LDA (OTLDA) of Huynh et al. (2020) is proposed to minimize the regularized optimal transport distance between document distribution $x_j$ and topic distribution about $\mathbf{M}$ in the vocabulary space. Also, Neural Sinkhorn Topic Model (NSTM) of Zhao et al. (2020) is proposed to directly minimize the OT distance between the topic proportion $\theta$ output from the encoder and the normalized BoW vector $\tilde{x}_j$. Compared with NSTM, by representing a document as a mixture of word embeddings and a mixture of topic embeddings, our model directly minimizes the CT cost between them in the same embedding space. Moreover, NSTM needs to feed the pretrained word embeddings to construct the cost matrix in Sinkhorn algorithm, while our WeTe can learn word and topic embeddings jointly from scratch. Finally, our model avoids the use of Sinkhorn iterations within each training iteration.

# 5 EXPERIMENTS

## 5.1 EXPERIMENTAL SETTINGS

**Datasets:** To demonstrate the robustness of our WeTe in terms of learning topics and document representation, we conduct the experiments on six widely-used textual data, including regular and short documents, varying in scales. The datasets include 20 News Group (20NG), DBpedia (DP) (Lehmann et al., 2015), Web Snippets (WS) (Phan et al., 2008), Tag My News (TMN) (Vitale et al., 2012), Reuters extracted from the Reuters-21578 dataset, and Reuters Corpus Volume 2 (RCV2) (Lewis et al., 2004), where WS, DP, and TMN consist of short documents. The statistics and detailed descriptions of the datasets are provided in Appendix C.

**Evaluation metrics:** Following Dieng et al. (2020) and Zhao et al. (2020), we use Topic Coherence (TC) and Topic Diversity (TD) to evaluate the quality of the learned topics. TC measures the average Normalized Pointwise Mutual Information (NPMI) over the top 10 words of each topic, and a higher score indicates more interpretable topics. TD denotes the percentage of unique words in the top 25

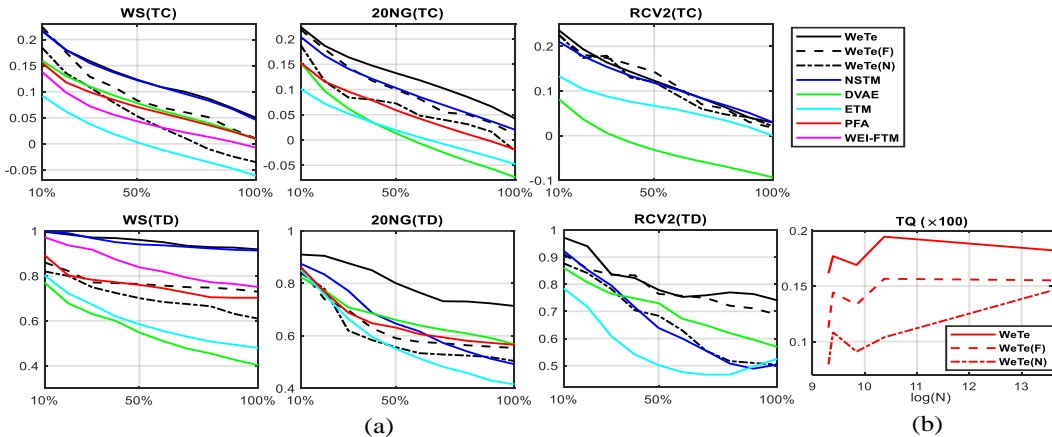

Figure 1: (a) The first and second rows show topic coherence (TC) and topic diversity (TD), respectively, for different methods on five datasets. In each subfigure, the horizontal axis indicates the proportion of selected topics according to their NPMIs. For both TC and TD, higher is better. (b) topic quality (TQ = TC ∗ TD) tendency of WeTe and its variants as the corpus size $N$ grows. Where, WeTe(F) and WeTe(N) denote that we finetune the word embeddings and learn it from scratch, respectively.

words of the selected topics. To comprehensively evaluate topic quality, we choose the topics with the highest NPMI and report the average score over those selected topics, where we vary the proportion of the selected topics from $10\%$ to $100\%$. Besides topic quality, we calculate Normalized Mutual Information (NMI) (Schütze et al., 2008) and purity on WS, RCV2, DP, and 20NG on clustering tasks, where we use the 6 super-categories as 20NG's ground truth and denote it as 20NG(6). We split all datasets according to their default training/testing division, and train a model on the training documents. Given the trained model, we collect the topic proportion $\theta$ on the testing documents and apply the K-Means algorithm on it, where the purity and NMI of the K-Means clusters are measured. Similar to Zhao et al. (2020), we set the number of clusters of K-Means as $N = 52$ for RCV2 and $N = 20$ for the other datasets. For all the metrics, higher values mean better performance.

**Baseline methods and their settings:** We compare the performance of our proposed model with the following baselines: **1)** traditional BPTMs, including **LDA** (Blei et al., 2003), a well-known topic model (here we use its collapsed Gibbs sampling extension (Griffiths & Steyvers, 2004)) and Poisson Factor Analysis (**PFA**) (Zhou et al., 2012), a hierarchical Bayesian topic model under the Poisson likelihood; **2)** VAEs based NTMs, such as Dirichlet VAE (**DVAE**) (Burkhardt & Kramer, 2019) and Embedded Topic Model (**ETM**) (Dieng et al., 2020), a generative model that marries traditional topic models with word embeddings; **3)** OT based NTM, Neural Sinkhorn Topic model (**NSTM**) (Zhao et al., 2020), which learns the topic proportions by directly minimizing the OT distance to a document's word distribution; **4)** TMs designed for short texts, including Pseudo-document-based Topic Model (**PTM**) (Zuo et al., 2016) and Word Embedding Informed Focused Topic Model (**WEI-FTM**) (Zhao et al., 2017a). In summary, ETM, NSTM, WEI-FTM, and our WeTe are the ones with pretrained word embeddings. For all baselines, we use their official default parameters with best reported settings.

**Settings for our proposed model:** Besides the default WeTe which loads the pretrained word embeddings from GloVe, we propose two variants of WeTe. The first variant initializes word embeddings from the Gaussian distribution $\mathcal{N}(0, 0.02)$ and learn word and topic embeddings jointly from the given dataset. The second variant loads the GloVe embeddings and fine-tunes them with other parameters. We denote those two variants as WeTe(N) and WeTe(F), respectively. We set the number of topics as $K = 100$. For our encoder, we employ a neural network stacked with a 3-layer $V$-256-100 fully-connected layer ($V$ is the vocabulary size), followed by a softplus layer. We set the trade-off hyperparameter as $\epsilon = 1.0$ and batch size as 200. We use the Adam optimizer (Kingma & Ba, 2015) with learning rate 0.001. All experiments are performed on an Nvidia RTX 2080-Ti GPU and implemented with PyTorch.

## 5.2 RESULTS

**Quantitative results:** For all models, we run the algorithms in comparison five times by modifying only the random seeds and report the mean and standard deviation (as error bars). We first examine

Table 1: Comparison of K-Means clustering purity (km-Purity) and NMI (km-NMI) for various methods. We use the 6 super-categories as 20NG's ground truth and denote it as 20NG(6). The best and second best scores of each dataset are highlighted in boldface and with an underline, respectively.

| Method | km-Purity(%) | | | | km-NMI(%) | | | |
|---|---|---|---|---|---|---|---|---|
| | WS | RCV2 | DP | 20NG(6) | WS | RCV2 | DP | 20NG(6) |
| LDA-Gibbs | 46.4±0.6 | 52.4±0.4 | 60.8 ±0.5 | 59.2±0.6 | 25.1±0.4 | 38.2±0.5 | 54.7 ±0.3 | 32.4 ±0.4 |
| PFA | 55.7±0.4 | - | 64.6 ±0.7 | 61.2±0.6 | 31.1±0.3 | - | 55.4±0.5 | 32.7 ±1.1 |
| PTM | 33.2 ±1.1 | - | 56.3 ±1.7 | - | 7.9±1.4 | - | 45.2 ±1.5 | - |
| WEI-FTM | 54.6±1.5 | - | 65.3 ±2.4 | - | 32.4±1.5 | - | 59.7±1.6 | - |
| DVAE | 26.6±1.5 | 52.6±1.2 | 67.2 ±1.1 | 64.6 ±1.0 | 3.7 ±0.8 | 31.3±0.9 | 50.8 ±0.6 | 29.8 ±0.6 |
| ETM | 32.9±2.3 | 50.2±0.6 | 63.1 ±1.5 | 62.6 ±2.2 | 12.3±2.3 | 30.3±1.0 | 53.2 ±0.7 | 29.3 ±1.5 |
| NSTM | 42.1±0.6 | 53.8±1.0 | 20.2 ±0.7 | 62.6±1.2 | 17.4 ±0.6 | 36.8±0.3 | 6.63±0.11 | 31.1 ±1.2 |
| WeTe | 59.0±0.1 | 59.2±0.2 | 75.8 ±0.8 | 67.3 ±0.6 | 34.5±0.1 | 40.3±0.4 | 62.5±0.8 | 35.0 ±0.4 |
| WeTe(N) | 59.7±0.1 | 58.5±0.3 | 74.1 ±3.3 | **70.2** ±1.0 | 34.1±0.1 | 41.2±0.1 | 60.1±1.1 | 34.3 ±0.8 |
| WeTe(F) | **60.8**±0.2 | **62.9**±0.5 | **77.1** ±1.0 | 68.5 ±0.2 | **34.9**±0.4 | **42.8**±0.3 | **63.7**±0.4 | **36.3** ±0.2 |

the quality of the topics discovered by WeTe. Fig. 1(a) shows the results of TC and TD on three corpora (more results can be found in Appendix D), varying in scales. Due to limited space, we only choose PFA and WEI-FTM as representatives of their respective methods. Since the Gibbs sampling based methods (*e.g.*, PFA, WEI-FTM) require walking through all documents in each iteration, it is not scalable to big data like RCV2. WEI-FTM only works on short texts. There are several observations drawn from different aspects. For the short texts (WS), WeTe has comparable performance with NSTM, and is much better than WEI-FTM, which is designed specifically for short texts. This observation confirms that our WeTe is effective and efficient in learning coherent and diverse topics from the short texts with pretrained word embeddings, without designing specialized architectures. In addition, for the regular and large datasets (20NG, RCV2), our proposed WeTe significantly outperforms the others in TC while achieving higher TD. Although some TMs (NSTM, ETM, WEI-FTM) utilize the pretrained word embeddings, it is demonstrated that how to assimilate them into topic model is the key factor. Thus we provide a reference for future studies along the line of combining word embeddings and topic models. Compared with WeTe, WeTe(F) and WeTe(N) need to learn word and topic embeddings from the current corpus, whose size is usually less than 1M, resulting in sub-optimal topics discovering. From Fig. 1(a), we can further find that those two variants achieve comparable result with other NTMs for top-20% topics, which means the proposed model has the ability to discover interpretable topics only from the given corpus without loading pretrained word embeddings. Fig. 1(b) denotes topic quality of our WeTe and its variants with different corpus scalar. it shows that WeTe(F) and WeTe(N) reach a performance close to that of WeTe as the scalar of the corpus becomes large, suggesting that the proposed model alone has the potential to learn meaningful word embeddings by itself on a large dataset.

The clustering Purity and NMI for various methods are shown in Table 1. Overall, the proposed model is superior to its competitors on all datasets. Compared with NSTM, which learns topic proportions $\theta$ by minimizing the OT distance to a document's word distribution, WeTe employs a probabilistic bidirectional transport method to learn $\theta$ and topic embeddings jointly, resulting in more distinguishable document representations. Besides, with the ability to finetune/learn word embeddings from the current corpus, WeTe(F) and WeTe(N) can better infer the topic proportion $\theta$, and hence give better performance in terms of document clustering. Those encouraging results show that not only the proposed model can discover the topics with high quality, but also learn good document representations for downstream clustering task on both short and regular documents. It thus indicates the benefit of minimizing the semantic distance between mixture of word embeddings and mixture of topic embeddings.

**Hyperparameter sensitivity** We fix the hyperparameter $\epsilon = 1.0$, which controls the weight of the Poisson likelihood in Eq. 13 in the previous experiments for fair comparison. Here we

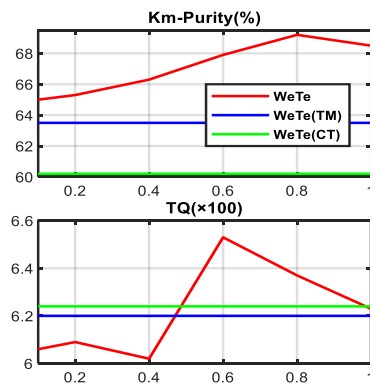

Figure 2: Parameter sensitivity of WeTe on 20NG dataset, KMeans clustering purity (Km-Purity) and Topic Quality (TQ).

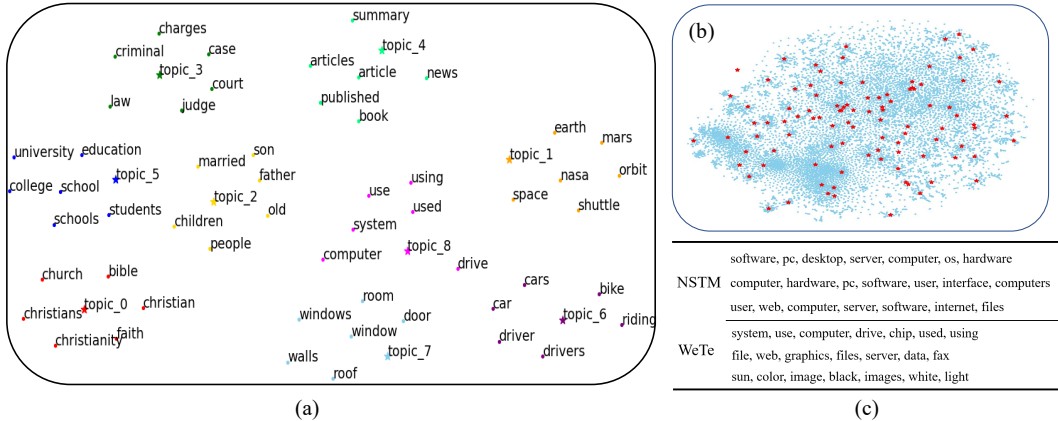

Figure 3: (a): t-SNE visualisation of selected topics and their top-6 words in the shared word embedding space. Different colors distinguish different topics; (b): Panoramic view of all words and learned topics; (c): Comparison of cherry-picked top-3 NSTM and WeTe topics on 20NG related to *desktop* keyword. In (a) and (b), stars and dots represent topic embeddings and word embeddings, respectively.

report the result of WeTe on 20NG with different $\epsilon$ in Fig. 2, where topic quality (TQ) is calculated as TQ $= $ TC $*$ TD. We also report two variants of WeTe, where one is trained using only the CT cost (WeTe(CT)), while the other using only likelihood (WeTe(TM)). We can see that 1), $\epsilon$ can be fine-tuned to balance betweeen document representation and topic quality. By carefully fine-tuning $\epsilon$ for each dataset, one can achieve even better performance than those reported in our experiments. 2), The CT cost leads to high topic quality, and the likelihood has benefits for the representation of documents. By combining these two objectives together, WeTe can produce better performance than using only either of them.

**Qualitative analysis:** Fig. 3(a) visualizes the learned topic embeddings. we present the top-9 topics with the highest NPMI learned by our proposed model on 20NG. For each topic, we select its top-6 words according to $\phi_k$, and then feed their embeddings together into the t-SNE tool (Van der Maaten & Hinton, 2008). We can observe that the topics are highly interpretable in the word embedding space, where each topic is close to semantically related words. Besides, those words under the same topic are closer together, and words under different topics are far apart. We can also see that the related topics are also closer in the embedding space, such as topic #2 and topic #5. Fig. 3(b) gives an overview of all word embeddings and learned topic embeddings. We find that the topic embeddings (red stars) are distributed evenly in the word embedding space, each of which plays the role of a concept center surrounded by semantically related words. Those interesting results illustrate our motivation that we can use the mixtures of topic embeddings to represent mixtures of word embeddings based on the CT cost between them. Given the query *desktop*, Fig. 3(c) compares three most related topics learned from NSTM and WeTe, where WeTe tends to discover more diverse topics than NSTM. We attribute this to the introduction of topic-to-doc cost, which enforces the topic to transport to all words that are semantically related to it with some probability. More qualitative analysis on topics are provided in Appendix E.

## 6 CONCLUSION

We introduce WeTe, a new topic modeling framework where each document is viewed as a bag of word-embedding vectors and each topic is modeled as an embedding vector in the shared word-embedding space. WeTe views the learning of a topic model as the process of minimizing the expected CT cost between those two sets over all documents. which avoids several challenges of existing TMs. Extensive experiments show that the proposed model outperforms competitive methods for both mining high quality topics and deriving better document representation. Thanks to the introduction of the pretrained word embeddings, WeTe achieves superior performance on short and regular texts. Moreover, the proposed model reduces the need to pre-define the size of the vocabulary, which makes WeTe more flexible in practical tasks.

## 7 ACKNOWLEDGEMENTS

H. Zheng, K. Tanwisuth, and M. Zhou acknowledge the support of NSF IIS-1812699 and a gift fund from ByteDance Inc. B. Chen acknowledges the support of NSFC (U21B2006 and 61771361), Shaanxi Youth Innovation Team Project, the 111 Project (No. B18039) and the Program for Oversea Talent by Chinese Central Government.

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

## A    DERIVATION OF EQUATION 12

The total CT cost can be written as:

$$
L = \frac{1}{J} \sum_{j=1}^{J} [L_{Q_j \to P_j} + L_{P_j \to Q_j}]
$$

$$
= \frac{1}{J} \sum_{j=1}^{J} [\sum_{k=1}^{K} \tilde{\theta}_{jk} \sum_{i=1}^{N_j} c(\boldsymbol{w}_{ji}, \boldsymbol{\alpha}_k) \pi(\boldsymbol{w}_{ji} \mid \boldsymbol{\alpha}_k) + \sum_{i=1}^{N_j} \frac{1}{N_j} \sum_{k=1}^{K} c(\boldsymbol{w}_{ji}, \boldsymbol{\alpha}_k) \pi(\boldsymbol{\alpha}_k \mid \boldsymbol{w}_{ji})],
$$

where

$$
\pi_{N_j}(\boldsymbol{w}_{ji} \mid \boldsymbol{\alpha}_k) = \frac{e^{-d(\boldsymbol{w}_{ji}, \boldsymbol{\alpha}_k)}}{\sum_{i'=1}^{N_j} e^{-d(\boldsymbol{w}_{ji'}, \boldsymbol{\alpha}_k)}}
$$

and

$$
\pi(\boldsymbol{\alpha}_k \mid \boldsymbol{w}_{ji}) = \frac{e^{-d(\boldsymbol{w}_{ji}, \boldsymbol{\alpha}_k)} \tilde{\theta}_{jk}}{\sum_{k'=1}^{K} e^{-d(\boldsymbol{w}_{ji}, \boldsymbol{\alpha}_{k'})} \tilde{\theta}_{jk'}}.
$$

Recall the definition of $c(\boldsymbol{w}_{ji}, \boldsymbol{\alpha}_k)$ in Equation 10 of the main paper, and $d(\boldsymbol{w}_{ji}, \boldsymbol{\alpha}_k) = -\boldsymbol{w}_{ji}^{\mathrm{T}} \boldsymbol{\alpha}_k$. With the fact that $\sum_{k=1}^{K} \tilde{\theta}_{jk} = 1$, we can rewrite the total CT cost $L$ as:

$$
L = \frac{1}{J} \sum_{j=1}^{J} [\sum_{k=1}^{K} \tilde{\theta}_{jk} \sum_{i=1}^{N_j} e^{-\boldsymbol{w}_{ji}^{\mathrm{T}} \boldsymbol{\alpha}_k} \frac{e^{\boldsymbol{w}_{ji}^{\mathrm{T}} \boldsymbol{\alpha}_k}}{\sum_{i'=1}^{N_j} e^{\boldsymbol{w}_{ji}^{\mathrm{T}} \boldsymbol{\alpha}_k}} + \sum_{i=1}^{N_j} \frac{1}{N_j} \sum_{k=1}^{K} e^{-\boldsymbol{w}_{ji}^{\mathrm{T}} \boldsymbol{\alpha}_k} \frac{e^{\boldsymbol{w}_{ji}^{\mathrm{T}} \boldsymbol{\alpha}_k} \tilde{\theta}_{jk}}{\sum_{k'=1}^{K} e^{\boldsymbol{w}_{ji}^{\mathrm{T}} \boldsymbol{\alpha}_k)} \tilde{\theta}_{jk'}}]
$$

$$
= \frac{1}{J} \sum_{j=1}^{J} \left[ \left( \sum_{k=1}^{K} \frac{\tilde{\theta}_{jk}}{\sum_{i'=1}^{N_j} e^{w_{ji'}^{T} \boldsymbol{\alpha}_k} \frac{1}{N_j}} \right) + \left( \sum_{i=1}^{N_j} \frac{\frac{1}{N_j}}{\sum_{k'=1}^{K} e^{w_{ji}^{T} \boldsymbol{\alpha}_{k'}} \tilde{\theta}_{jk'}} \right) \right]
$$

which, to the best of our knowledge, does not resemble any existing topic modeling loss functions. The two terms in the bracket have a very intriguing relationship, where in the fraction formula $\tilde{\theta}_{jk}$ and $\frac{1}{N_j}$ swap their locations and $\sum_k$ and $\sum_i$ also swap their locations. To minimize the first term, we will need to ensure the denominator $\sum_{i'=1}^{N_j} e^{w_{ji'}^{T} \boldsymbol{\alpha}_k} \frac{1}{N_j}$ to be sufficiently large whenever $\tilde{\theta}_{jk}$ is non-negligible, which can be achieved only if the inner products of the words in document $j$ and topic $k$ aggregate to a sufficiently large value whenever $\tilde{\theta}_{jk} > 0$ (*i.e.*, each inferred topic embedding vector needs to be close to at least one word embedding vector of a given document when that topic has a non-negligible proportion in that document). To minimize the second term, we will need to ensure the denominator $\sum_{k'=1}^{K} e^{w_{ji}^{T} \boldsymbol{\alpha}_{k'}} \tilde{\theta}_{jk'}$ to be large for every single word, which for word $i$ can be achieved only if there exists at least one topic that has a large inner product with word $i$ (*i.e.*, each word can find at least a single non-negligibly-weighted topic that is sufficiently close to it, in other words, the inferred topics need to have a good coverage of the word embedding space occupied by the corpus).

## B    TRAINING ALGORITHM

The training procedure of the proposed WeTe is shown in Algorithm 1.

## C    DATASETS

Our experiments are conducted on six widely-used benchmark text datasets, varying in scales and document lengths, including 20 News Group (20NG), DBpedia (DP) (Lehmann et al., 2015), Web Snippets (WS) (Phan et al., 2008), Tag My News (TMN) (Vitale et al., 2012), Reuters extracted from the Reuters-21578 dataset, and Reuters Corpus Volume 2 (RCV2) (Lewis et al., 2004), where WS, DP, and TMN consist of short documents. To demonstrate the scalability of the proposed model for document clustering, For multi-label RCV2 dataset, we left documents in test dataset with single label at second level, resulting in 52 categories. We load the pretrained word embeddings from GloVe[1] (Pennington et al., 2014).

---

[1] https://nlp.stanford.edu/projects/glove/

---

**Algorithm 1** Training algorithm for our proposed model.

---

**Input**: training documents, pretrained word embeddings $\mathbf{E}$, topic number $K$, hyperparameter $\epsilon$.
**Initialize**: topic embeddings $\boldsymbol{\alpha}$, encoder parameters $\mathbf{W}$.
**for** iter = 1,2,3,... **do**
  Sample a batch of $J$ input documents and represent them as the empirical distributions $\{P_j\}_{j=1}^J$
  and form the document-specific empirical topic distribution $\{Q_j\}_{j=1}^J$;
  With the cost function in Equation 10 and transport probabilities in Equation 9 and Equation 6,
  compute the CT loss with Equation 12 as the first term of Equation 13;
  Compute the topic $\mathbf{M}$ with Equation 11 and the topic proportions $\{\boldsymbol{\theta}_j\}$ with input $\boldsymbol{x}_j$, denoted
  as $q(\boldsymbol{\theta}_j|\boldsymbol{x}_j) = \text{Weibull}(f_{\boldsymbol{W}}(x_j), g_{\boldsymbol{W}}(x_j))$; compute the second term of Equation 13;
  Update $\boldsymbol{\alpha}$ and $\mathbf{W}$ according to Equation 13;
**end for**

---

Table C. 1: Statistics of the datasets

|        | Number of docs | Vocabulary size(V) | average length | Number of labels |
|--------|----------------|--------------------|----------------|------------------|
| 20NG   | 18,864         | 22,636             | 108            | 6                |
| DP     | 449,665        | 9,835              | 22             | 14               |
| WS     | 12,337         | 10,052             | 15             | 8                |
| TMN    | 32,597         | 13,368             | 18             | 7                |
| Reuters| 11,367         | 8,817              | 74             | N/A              |
| RCV2   | 804,414        | 7,282              | 75             | 52               |

- **20NG**[2]: 20 Newsgroups consists of 18,846 articles. We remove stop words and words with document frequency less than 100 times. We also ignore documents that contain only one word from the corpus. We only use the 6 super-categories as 20NG's ground truth and denote it as 20NG(6) in the clustering task, as there are confusing overlaps in its official 20 categories, e.g., *comp.sys.ibm.pc.hardware* and *comp.sys.mac.hardware*.

- **DP**[3]: DBpedia is a crowd-sourced dataset extracted from Wikipedia pages. We follow the pre-processing process in Zhang et al. (2015), where the fields that we use for this dataset contain title and abstract of each Wikipedia article.

- **WS**: Web Snippets, used in Li et al. (2016) and Zhao et al. (2020), contains 12,237 web search snippets with 8 categories. There are 10,052 tokens in the vocabulary and the average length of a snippet is 15.

- **TMN**[4]: Tag My News consists of 32,597 RSS news snippets from Tag My News with 7 categories. Each snippet contains a title and a short description, and the average length of a snippet is 18.

- **Reuters**[5]: A widely used corpus extracted from the Reuters-21578 dataset. We only use it on topic quality task, and there are 11,367 documents with 8,817 tokens in vocabulary.

- **RCV2**[6]: Reuters Corpus Volume 2, used in Zhao et al. (2020), consists of 804,414 documents, whose vocabulary size is 7282 and average length is 75.

A summary of dataset statistics is shown in Table C. 1.

## D ADDITIONAL TOPIC QUALITY RESULT

In Fig. D. 1, we report topic coherence (TC) and topic diversity (TD) for varied methods on TMN and Reuters dataset, which confirms that our proposed model outperforms the others in high quality topic discovering.

---

[2]http://qwone.com/ jason/20Newsgroups
[3]https://en.wikipedia.org/wiki/Main_Page
[4]http://acube.di.unipi.it/tmn-dataset/
[5]https://kdd.ics.uci.edu/databases/reuters21578/reuters21578.html
[6]https://trec.nist.gov/data/reuters/reuters.html

When the topic number becomes insufficient, the topic distribution $p(\boldsymbol{m}_k \,|\, k)$ often resembles the corpus distribution $p(w)$, where high frequency words become the top terms related to most topics. We want topics learned from WeTe to be specific (*e.g.*, not overly general). Topic Specificity (TS) is defined by the average KL divergence from each topic's distribution to the corpus distribution:

$$TS = \frac{1}{K} \sum_{k=1}^{K} KL(p(\boldsymbol{m}_k \,|\, k) || p(w)).$$

Jointly with topic diversity and topic coherence, we report topic specificity (TS) of various methods on six datasets in Table D. 1. it can be found that the proposed model is superior to its competitors on all datasets, indicating that WeTe produces more useful and specific topics than other NTMs.

In Tables D. 2, D. 3, and D. 4, we show the top-10 words of the selected topics learned from WeTe and its two variants on 20NG, TMN, and RCV2, respectively. We note that the proposed model can not only learn meaningful topics from the pretrained word embeddings, but also learn word and topic embeddings jointly from scratch, discovering equally meaningful topics.

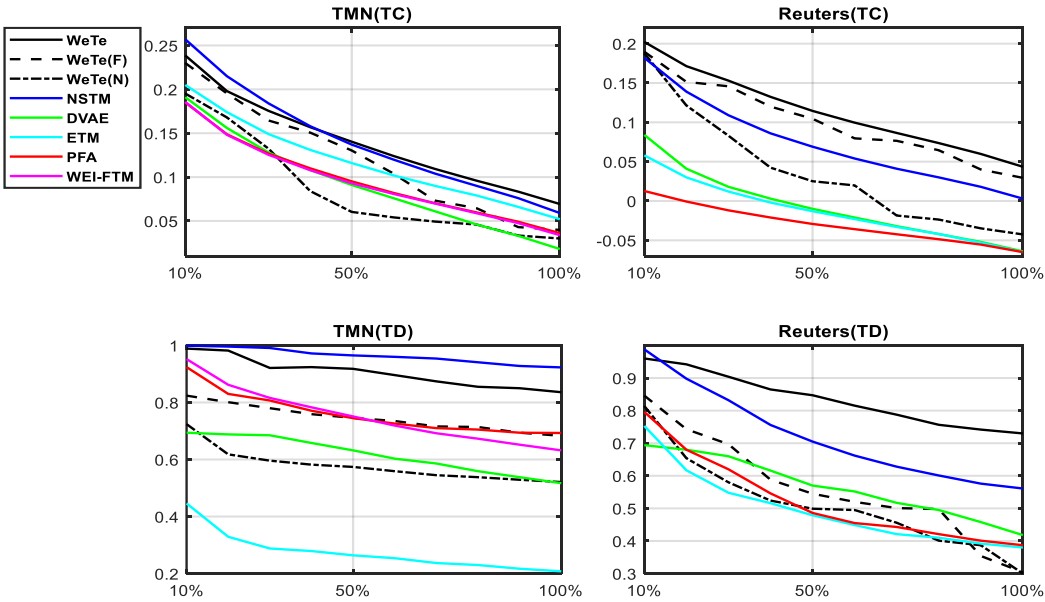

Figure D. 1: The first and second rows show topic coherence (TC) and topic diversity (TD), respectively, for different methods on the TMN and Reuters datasets. In each subfigure, the horizontal axis indicates the proportion of selected topics according to their NPMIs. For both TC and TD, higher is better. WeTe(F) and WeTe(N) denote that we finetune the word embeddings and learn it from scratch, respectively.

Table D. 1: Topic specificity (TS) of various methods on web(WS), 20NG, DP, RCV2, TMN, and Twitter datasets, higher is better.

| Method | WS | 20NG | DP | RCV2 | TMN | Twitter |
|---|---|---|---|---|---|---|
| LDA | 3.84 | 4.67 | 5.42 | 7.08 | 3.89 | 3.95 |
| DVAE | 2.50 | 3.12 | 4.04 | 5.45 | 2.86 | 1.73 |
| NSTM | 1.49 | 1.97 | 4.47 | 6.24 | 1.07 | 2.27 |
| WeTe | **4.51** | 5.71 | 5.58 | **7.94** | **4.16** | **4.43** |
| WeTe(F) | 4.48 | **5.74** | **5.74** | 7.42 | 4.07 | 4.36 |
| WeTe(N) | 4.01 | 5.42 | 5.38 | 6.98 | 3.89 | 4.13 |

Table D. 2: Topics learned from WeTe, WeTe(F), and WeTe(N) on the 20NG dataset, where the top-10 words for each topic are visualized.

| Method | Top words |
|--------|-----------|
| WeTe | space nasa orbit spacecraft mars shuttle launch flight rocket solar
window image display color screen graphics output motif mode format
game team hockey nhl play teams players win player league season |
| WeTe(F) | space satellite launch nasa shuttle mission research lunar earth technology
window problem card monitor mouse video windows driver memory screen
team hockey game players season league play goal year teams |
| WeTe(N) | space launch satellite nasa shuttle earth lunar first mission system
window display application server mit screen problem use get program
year game team players baseball runs games last season league |

Table D. 3: Topics learned from WeTe, WeTe(F), and WeTe(N) on the TMN dataset, where the top-10 words for each topic are visualized.

| Method | Top words |
|--------|-----------|
| WeTe | million billion company buy group share amp firm bid sell
wedding idol royal william prince singer star kate rock taylor
team season sports league teams soccer field manchester briefing club |
| WeTe(F) | million billion deal group company firm offer buy shares sell
star stars movie fans idol hollywood box fan film super
players nfl coach draft teams football basketball player nba lockout |
| WeTe(N) | million company video deal online internet apple google mobile media
show star theater book idol royal dies space wedding music
coach nfl players team state season sports national tournament basketball |

Table D. 4: Topics learned from WeTe, WeTe(F), and WeTe(N) on the RCV2 dataset, where the top-10 words for each topic are visualized.

| Method | Top words |
|--------|-----------|
| WeTe | million total billion asset worth sale cash debt cost payout
oil gas fuel barrel palm petroleum gulf shell bpd cubic olein
network internet custom access microsoft web design tv broadcast media |
| WeTe(F) | sale sold bought sell retail buy chain auction supermarket shop discount
oil barrel nymex brent gas petroleum fuel gallon wti gulf
system network personnel microsoft inform chief internet web unit custom |
| WeTe(N) | percent billion year million market rate month economic growth dollar
oil gas barrel brent fuel output sulphur petroleum nymex diesel gallon
network channel radio tv media station broadcast film video disney |

## E   THE LEARNED WORD EMBEDDINGS

WeTe(N) provides a new method to learn word embeddings from scratch. Recall the topic-to-doc CT cost for a special document $j$ in WeTe:

$$C_j = c(\boldsymbol{w}_{ji}, \boldsymbol{\phi}_k) \frac{e^{-d(\boldsymbol{w}_{ji}, \boldsymbol{\phi}_k)}}{\sum_{i'=1}^{N_j} e^{-d(\boldsymbol{w}_{ji'}, \boldsymbol{\phi}_k)}}, \quad \boldsymbol{w}_{ji} \in \{\boldsymbol{w}_{j1}, \dots, \boldsymbol{w}_{jN_j}\}.$$

This transport cost mirrors the likelihood in skip-gram model. Such skip-gram models use the central word to predict the surrounding words. By contrast, our WeTe uses the topic embedding vectors $\phi$ as the central words and generates the document words, rather than a window of surrounding words. In other words, skip-gram models can be viewed as a special variant of WeTe with the window size $c = N_j$. To evaluate the word embeddings learned from WeTe(N), given a query word, we visualize top-8 words that are most closest to it. We compare WeTe(N) with GloVe in Table E. 1.

Compared to GloVe, the word embedding learned by WeTe(N) tends to be more semantically diverse. For example, "download", "modem" for "pc", and "goal", "win" for "game". We attribute this to the use of the document-level context.

Table E. 1: Comparison of the most relevant words for the query words on RCV2 dataset.

| Query word | Method | Top words |
|---|---|---|
| **pc** | GloVe | pc desktop computer software macintosh computers pentium pcs microsoft xp |
| | WeTe(N) | pc desktop macintosh pcs microsoft internet os download mac modem |
| **game** | GloVe | game games season play match player league team scored playoffs |
| | WeTe(N) | game season play match team playoff bowl goal win coach |
| **world** | GloVe | world cup international olympic european championships event europe |
| | WeTe(N) | world cup international european event asian asia women nation team |
| **school** | GloVe | school college university schools students education elementary graduate |
| | WeTe(N) | school high student campus district church program degree taught harvard |

## F  COMPLEXITY ANALYSIS

Table F. 1: Time and space complexity analysis of WeTe. CT and TM denote the conditional transport part and topic model part, respectively. We ignore the 3-layer encoder because it is shared with all neural topic models.

| | CT | TM |
|---|---|---|
| Time complexity | $(2V+4N_B)K$ | $2VKd^2 + BVKK^2 + 4VB$ |
| Space complexity | $(V+K)d + (2V+4N_B)K$ | $(V+B)K+VB$ |

As a neural topic model, WeTe has a comparable complexity to other neural topic models. In detail, for a mini-batch of documents with batch-size $B$, $N_B$ denotes the total words in the mini-batch. We summary the time and space complexity at Table F. 1, where CT denotes conditional transport and TM means the topic model, we here ignore the 3-layer neural encoder, due to it is shared with other neural topic models. $V$ is the vocabulary size, $K$ is the number of topics and $d$ is the embedding size. We can see that CT obtains linear complexity in both time and space with respect to the vocabulary and the total number of words in the mini-batch.

We also compare WeTe with other three NTMs on large RCV2 (V=13735,N=804,414) with a large topic setting (K=500). All the methods are evaluated on an Nvidia RTX 2080-Ti GPU with batch size of 500. The normalized training loss is shown in Fig. F. 1, where the direct comparability between losses is not available due to the different designs. It demonstrates that the proposed model has acceptable learning speed compared with other NTMs. Fig. F. 1(a) shows that WeTe requires fewer iterations compared to DVAE and ETM. And Fig. F. 1(b) demonstrates that our WeTe has similar time consumption to DVAE. Although ETM and NSTM have faster training speed, their performance on both topic quality and clustering task is incomparable to ours. In other words, WeTe balances the performance and speed well.

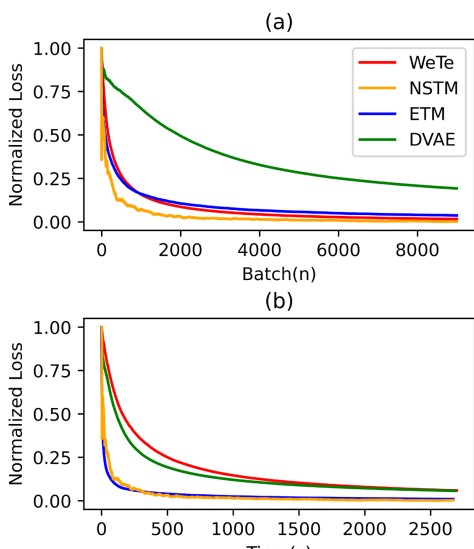

Figure F. 1: Training loss on RCV2 over batches (a) and seconds (b).

