# OpenReview forum: "Representing Mixtures of Word Embeddings with Mixtures of Topic Embeddings"
_ICLR.cc/2022/Conference — ICLR 2022 Poster_

### Official Review · Reviewer_Bvas · 2021-10-16

**Correctness:** 3
**Technical Novelty And Significance:** 2
**Empirical Novelty And Significance:** 3
**Recommendation:** 5
**Confidence:** 5

**Main Review:**

Strengths:
1. A diversity of datasets including web pages and news articles are used in the evaluation, and experiments show the superiority of the proposed topic modeling method.
2. The algorithm is reasonable and has the advantage that it can learn word embeddings and topic embeddings from scratch.
3. The paper is well written and easy to follow. The authors provide concrete figures and tables to show the results of experiments.

Weaknesses:
1. Given a few recent works on learning dense word vectors jointly with latent document-level mixtures of topic vectors, the novelty of this paper is not significant. For example, there's a missing reference:  Moody, Christopher E. "Mixing dirichlet topic models and word embeddings to make lda2vec." arXiv preprint arXiv:1605.02019 (2016).

2. The experiment results shown in this paper do not include t-test results such as p-values. We cannot tell how significant the performance of the proposed algorithm is, compared to other state-of-the-art methods.

3. To show the interpretability of the resulting topics, it'd be great if the authors can provide qualitative analysis such as in Chang, Jonathan, et al. "Reading tea leaves: How humans interpret topic models." Advances in neural information processing systems. 2009.

4. For comparison of time/space complexities, it'd be great if the authors can provide a detailed complexity analysis.




**Summary Of The Paper:**

This paper presents a new topic modeling framework called WeTe: each document in text corpus is represented as a bag of word embeddings vectors, and each topic is modeled as an embedding vector in the shared word embedding space. WeTe minimizes the expected difference between those two sets over all documents. A bidirectional transport-based method is proposed to learn the topic embeddings as well as topic proportions for documents efficiently. Extensive experiments on news and web pages show that the proposed model outperforms competitive methods for both deriving high-quality latent topics and generating better document representations for clustering tasks.

**Summary Of The Review:**

The idea of learning latent topics for text documents using bag-of-words embeddings is not quite new given that a few previous papers describe similar methods. However, it's still empirically useful to introduce concrete optimization methods to learn such a model. This paper presents extensive experiments on news and web pages to show the superiority of the proposed method, compared to other state-of-the-art methods.

---

> ### Author Response · Authors · 2021-11-22
> **Response to Reviewer Bvas**
>
> Thank you for your time and detailed feedback. The relevant missing references have been addressed in the revised version, with the major changes marked in blue. Below we respond to your questions/criticisms.
>
> **Q1**
>
> Thanks for the reference. We agree with you that learning dense word vectors jointly with latent document-level mixtures of topic vectors is related to our work and **we have discussed lda2vec in the related work in our revised version**. But there are fundamental differences between lda2vec and ours. Specifically, using embedding representations of both words and topics, WeTe formulates the topic model as the conditional transport problem where each document is viewed as a set of word embeddings. To discover those topic embeddings, WeTe aims to minimize the total bi-directional cost between the set of words embeddings and a set of global topic embeddings. This distinguishes our model from other likelihood-based or skipgram-base methods.
>
> Recall that the Skipgram Negative Sampling Loss (SGNS) in lda2vec can be denoted as:
>
> \begin{equation*}
>    \mathcal L_{ij}^{net} = log \sigma(c_j \cdot w_i) + \sum_{l=0}^{n} log \sigma(-c_j \cdot w_l)
> \end{equation*}
>
> Substituting $c_j$ with topic embeddings, one can obtain:
> \begin{equation*}
>     \mathcal L_{ij}^{net} = \text{log} \sigma(w_j \cdot w_i + (p_{j0} \cdot t_0 + p_{j1} \cdot t_1 + ... + p_{jn}\cdot t_n) \cdot w_i) + \sum_{l=0}^{n} \text{log} \sigma(-c_j \cdot w_l)
> \end{equation*}
> We are interested in the term $I = (p_{j0} \cdot t_0 + p_{j1} \cdot t_1 + ... + p_{jn}\cdot t_n) \cdot w_i = \sum_{n} p_{jn} \cdot t_n \cdot w_i$, where $I$ can be viewed as the doc-to-topic cost in WeTe: $C(\alpha, w_{ji}) = \sum_{k} \pi(\alpha_k|w_{ji}) c(\alpha_k, w_{ji})$. But different from lda2vec that directly uses the softmax to transport the document to its topic proportions $p_j$, WeTe calculates it by combining the topic proportions with the similarities between the words and topics:
> $\pi( \alpha_k | w_{ji}) = \frac{e^{-d(w_{ji},\alpha_k)} \tilde \theta_{jk}}{\sum_{k'=1}^K e^{-d(w_{ji},\alpha_{k'})}\tilde \theta_{jk'}}$,
> where the semantic information of words and topics in embedding space is considered again. Besides, WeTe also employs a topic-to-doc transport which forces the inferred topics are close to at least one word embedding vector of a given document. The transport of these two directions improved the learning of coherent topic discovering and document representation, which is lda2vec not considered fully.
>
> **Q2**
>
> Following your advice, we calculate p-value when comparing our WeTe with baselines, and report them below. We can see that all p-values for both Purity and NMI are far less than 0.05. This indicates that our WeTe indeed outperforms other competitive baselines.
>
> |   |  LDA   | PFA | DVAE | ETM | NSTM |
> |:---:|:---:|:---:|:----:|:---:|:----:|
> | p-value(Purity) | 7.98$e^{-9}$|5.44$e^{-9}$ |2.69$e^{-5}$| 3.33$e^{-4}$   |   4.16$e^{-6}$   |
> | p-value(NMI) | 4.97$^{-8}$ |9.24$^{-5}$|1.36$e^{-8}$ |6.59$e^{-6}$   |   1.18$e^{-5}$   |
>
> **Q3**
>
> Thanks for pointing out the paper of "Reading tea leaves: How humans interpret topic models", which is the fundamental one for evaluating topic quality for topic models.
> However, this method relies on manual annotation and has never been automated. To address this issue, people have proposed several automated methods that approximate the evaluation of the paper. As discussed in [1], topic coherence (TC) is the most popular one for doing so, which has been reported in our paper. TC can be viewed as an automated version of the paper of "Reading tea leaves: How humans interpret topic models".
>
> In addition to TC, we also report other metrics on topic quality, including topic diversity (TD), topic specificity (TS)
> and qualitative analysis including T-SNE visualization of topci embeddings. We believe that our evaluations on topic quality are very comprehensive.
>
> [1] Lau, Jey Han, David Newman, and Timothy Baldwin. "Machine reading tea leaves: Automatically evaluating topic coherence and topic model quality." EACL 2014.
>
> **Q4**
>
> As a neural topic model, WeTe has a comparable complexity to other neural topic models. In detail, for a minibatch of documents with batch-size $B$, $N_B$ denotes the total words in the minibatch. We summary the time and space complexity as below:
>
> |    | CT  | TM    |
> |:-------:|:---:|:-----:|
> | Time complexity  | (2V+4$N_B$)K      | 2VK$d^2$+BV$K^2$+4VB |
> | Space complexity | (V+K)d + (2V+4$N_B$)K | (V+B)K+VB           |
>
> where, CT denotes conditional transport and TM means the topic model, we here ignore the 3-layer neural encoder, due to it is shared with other neural topic models. $V$ is the vocabulary size, $K$ is the number of topics and $d$ is the embedding size.
> We can see that CT obtains linear complexity in both time and space with respect to the vocabulary and the total number of words in the minibatch.
>
> We have added the above result in appendix F in the revised version.

---

### Official Review · Reviewer_W5Cw · 2021-11-01

**Correctness:** 3
**Technical Novelty And Significance:** 3
**Empirical Novelty And Significance:** 3
**Recommendation:** 6
**Confidence:** 4

**Main Review:**

Strength of the paper : the point of view is relatively original, since it provides a new framework for topic modelling. It relies on a version of bidirectional optimal transport which is quite origineal and embedd words and topics in the same space. The relevance of the model is assess by numerical experiments comparing in terms of perplexity this approach with others ones more classical as vanilla LDA, OTLDA...

Weaknesses : the idea of mixing OT and LDA is not so new, even if bidirectional OT has not been used yet. In addition, this version of bidirectional OT is just a symmetrisation, which is not so original

**Summary Of The Paper:**

The aim of the paper is to revisit topic modeling through the lens of word embeddings and using a bidirectional version optimal transport. TIn this paper each document is considered as a set of embeddings words, each topic is seen as a mixture of embedding living in the same space as words embeddings
The aim is to minimize the distance of the topic distribution and the words distribution for each documents

**Summary Of The Review:**

The paper mixes OT and LDA in a rather original way. Nevertheless, this idea is not new and has been already proposed in other papers

---

> ### Author Response · Authors · 2021-11-22
> **Response to Reviewer W5Cw**
>
> Thank you for your time and the positive feedback.
>
> We agree with the point that the idea of mixing OT and LDA is not new. For example, existing models [1][2], which connect topic modelling with OT, usually adapt the idea of Wasserstein barycentres and Wasserstein Dictionary Learning. However, we aim to approximate each empirical distribution from document with a corresponding to-be-learned document-specific distribution, which does not belong the barycentre problem and provides a new view of topic modelling. Besides, while our WeTe is most relevant to the recent OT-based neural topic model NSTM, there are several key differences between ours and NSTM in terms of
> the relations between documents, topics, and words:
>
> i) NSTM minimizes the OT distance between the topic proportion $z$ and the normalized document bag-of-word representation $\widetilde{x}$. However, in WeTe, each document is viewed as a set of word embedding vectors and also a set of topic embedding vectors, where word embedding and topic embedding live in the same space. Naturally, WeTe views the learning of a topic model as the process of minimizing the conditional transport difference between those two sets over all documents.
>
> ii) In NSTM, to compute the optimal transport distance, one needs to perform the Sinkhorn algorithm, which involves a complex iterative optimization process in each mini-batch for each epoch. With the help of conditional transport, we can avoid such iterative process and directly optimize model with SGD.
>
> iii) Due to the conditional transport, we can parameterize the two conditional probabilities (Eq. 6 and 9) in a flexible way, as well as the cost function. Such flexibility offers better model capacity of ours over NSTM. Therefore, our WeTe outperforms NSTMs significantly on almost all datasets over various metrics.
>
> [1] Hongteng Xu, Wenlin Wang, Wei Liu, and Lawrence Carin. Distilled wasserstein learning for wordembedding and topic modeling. In NeurIPS2018.
>
> [2] Viet Huynh, He Zhao, and Dinh Phung. OTLDA: A geometry-aware optimal transport approach fortopic modeling. In NeurIPS2020.

---

### Official Review · Reviewer_2jw3 · 2021-11-01

**Correctness:** 3
**Technical Novelty And Significance:** 2
**Empirical Novelty And Significance:** Not applicable
**Recommendation:** 5
**Confidence:** 4

**Main Review:**

Strengths:
1. The authors develop a novel model to jointly model latent topics and word embeddings. While the model is novel, there are already existing works that have jointly modelled latent topics and word embeddings exploiting their complementary advantages. The authors also present the derivation of the model and the key equations that demonstrate how the model works.
2. The authors have also tried to intertwine their model with the experiments. As a result, the authors have conducted some experiments to demonstrate that the model improves upon some existing methods.

There are several ways in which this paper can be improved:
1. The authors can strengthen their experiments by experimenting against some downstream applications such as document classification. Unsure of why the authors did not cite this work that has used document classification as one of the experiments (https://dl.acm.org/doi/10.1145/3077136.3080806).
2. Relating to the work cited in (1) above, it would be nice to have some cooperation on how the proposed work differs from the work that is already published by Bei et al. There are some follow-up works too since this work was published and the authors are requested to have a go at them too. Where Bei's work would be crucial is mainly in comparing against the run-time performance of the models where Bei's work is a much simpler version of a model that jointly learns latent topics and embeddings. Besides that, the effectiveness of the model could be compared as a result.
3. Correcting some grammar issues, e.g., facilitates instead of facilities, latter instead of later; authors can use the PDF search feature to find such words.
4. The authors could consider theoretically comparing their embeddings with those static embeddings obtained using Skipgram or GloVe. Or, do we expect that the embeddings that the authors learn using their model are different in property from traditional static embeddings such as GloVe or Skipgram.

**Summary Of The Paper:**

Jointly learning latent topic representations with word embeddings has caught some attention in the past. The reason why researchers choose this route is to exploit the complementary advantage of both these models, for instance, to improve the performance of the model on short texts. In this paper, the authors study a new topic model where topics and words are encoded in the same embedding space. Usually, in the traditional topic modelling literature, the posterior inference is realised using algorithms such as Gibbs sampling or collapsed Gibbs sampling; however, in this paper, since the authors exploit complementary knowledge between the two models, the end-to-end model is jointly trained the document-specific discrete latent topic distribution is made to appear as close as possible to the discrete distribution over words in the document. A probabilistic bidirectional transport model is developed that measures the difference between the discrete distributions in the embedding space. The authors also discuss several key advantages of the model.

Experiments are then followed with topic diversity and coherence measures as evaluation metrics which are widely used in the literature. Qualitatively, the authors show some example results from their dataset in the form of a t-SNE plot.

**Summary Of The Review:**

While there are strengths in terms of the model novelty, there is some weakness too. The paper can be further improved from various angles such as strengthening the experiments section by comprising with some closely relevant works including conducting some downstream application analysis such as document classification performance.

---

> ### Author Response · Authors · 2021-11-22
> **Response to Reviewer 2jw3 (1/2)**
>
> Thank you for your time and the detailed feedback. The typos and relevant missing references have been addressed in the revised version, with the major changes marked in blue.  Below we respond to your questions/criticisms.
>
> **Q1**
>
> To our best knowledge, both document clustering and document classification can reflect the ability of learned document representation, and either one is sufficient for evaluating a topic model in terms of document representation. In STE, they only report the document classification results without document clustering results. In our submitted version, we follow the recent related work NSTM [1] and consider the document clustering task, where we had reported Purity and NMI on document clustering task at Table. 1. Results show that our WeTe outperforms other neural topic models. It indicates that the representations  derived from our model are more distinguishable. Following your suggestions, we also compare the proposed model with other neural topic model (e.g., ETM, DVAE, NSTM) on 20NG(6) dataset by considering document classification task, and report the accuracy score below. From the additional classification task, we can see that WeTe outperforms other NTMs, which is in accord with the phenomenon on the clustering task. If necessary, we will complete the classification tasks on other datasets in the final version. We couldn't compare with STE because they didn't publish the code.
>
> | Method |  ETM  |  DVAE |  NSTM |  WeTe |
> |:------:|:-----:|:-----:|:-----:|:-----:|
> |   ACC  | 79.52 | 70.65 | 83.36 | 85.42 |
>
> [1] He Zhao, D Phung, V Huynh, T Le, and W Buntine. Neural topic model via optimal transport, 2020
>
> **Q2**
>
> Following your advice, we have carefully gone through STE, and its follow-up works. In the revised version, **we have discussed the differences and connections between ours with STE in the related work Section.**
>
> Overall, our WeTe directly views topics as the to-be-learned embedding vectors. While STE assumes that a word $w$ may have a different representation under each topic, and thus implicitly models topics by viewing each word $w$ as a matrix $U_w \in \mathbb{R}^{K \times s}$ ($K$ is the number of topics and $s$ word embedding size).
> This difference causes the two models are different in many ways:
>
> 1) WeTe can calculate the transport probabilities between the global topics and words within a document via the similarities between their embedding vectors, while STE predicts the surrounding words $w_{t+j}$ of the central word $w_t$ under the given topic assignment via the similarities between the corresponding topical word embedding of two words, leading to an estimation sampling (negative sampling scheme);
>
> 2) In STE, the EM algorithm is employed to iteratively update topic proportion and word embedding matrix alternatively, unlike the end-to-end training in our WeTe;
>
> 3)  WeTe can visualize words and topics in the same embedding space, and explain topics by their top-n words calculated simply by the inner product of the word and topic embeddings. However, STE generates topics as the ranked list of bi-grams by traversing all bi-grams.
>
> There are several interesting connections between STE and our WeTe in some case. Recall that STE predicts the surrounding words by the central word. In our paper, the topic-to-doc transport in WeTe denotes the topic embedding as the central vector, and views words within a document as the surrounding words. That is to say, STE has a similar transport formation with the topic-to-doc transport in WeTe when the window size is set as the length of the document.
> Besides, the topic-to-doc cost in WeTe of a document $j$ is:
> \begin{equation*}
>     C = \sum_{i=1}^{N_j} \sum_{k=1}^{K} c(w_{ji}, \alpha_k) \frac{exp(-d(w_{ji},\alpha_k))}{\sum_{i'=1}^{N_j}exp(-d(w_{ji'},\alpha_k))}
> \end{equation*}
> Where $c(\cdot)$ is the cost function, and $d(\cdot)$ is the distance function. When we specify a "hard" function for c, for example when $w_{ij}\alpha_k>\epsilon$, $c=1$, otherwise, $c=0$, where $\epsilon$ is the threshold. Then we can obtain:
> \begin{equation*}
>     p(w_{ji}|\alpha,k) := C_{w_{ji},k} = \frac{exp(-d(w_{ji},\alpha_k))}{\sum_{i'=1}^{N_j}exp(-d(w_{ji'},\alpha_k))}
> \end{equation*}
> The above derivation mirrors the likelihood in STE. Therefore, STE can be viewed as a special variant of WeTe when ignoring the doc-to-topic cost and considering above relaxation .
>
> We summary the time and space complexity below, where, CT denotes conditional transport and TM means the topic model, $V$ is the vocabulary size, $K$ is the number of topics and $d$ is the embedding size.  We find that CT obtains linear complexity in both time and space with respect to the vocabulary and the total number of words in the minibatch
>
> | |CT|TM|
> |:--:|:--:|:--:|
> |Time complexity|(2V+4$N_B$)K|2VK$d^2$+BV$K^2$+4VB |
> |Space complexity|(V+K)d + (2V+4$N_B$)K|(V+B)K+VB  |
>
> We have added the above result in appendix F in the revised version.

---

> > ### Author Response · Authors · 2021-11-22
> > **Response to Reviewer 2jw3 (2/2)**
> >
> > **Q3**
> >
> > Thank you! We have made the corrections to improve the overall quality of our article.
> >
> > **Q4**
> >
> > As discussed in **Q1**, WeTe shares some similar properties with skip-gram. The base skip-gram formulation defines $p(w_{t+j}|w_t)$ using the softmax function:
> > \begin{equation*}
> >     p(w_O|w_I) = \frac{exp(v_{w_O}^{T} v_{w_I})}{\sum_{v=1}^{V}exp(v_{v}^T v_{w_I})}
> > \end{equation*}
> > The skip-gram model is similar with the topic-to-doc transport in WeTe (ignoring the doc-to-topic transport), where the topic vectors are viewed as the central words, and the words within a document are viewed as its surrounding words. By setting the window size $c=N_j$, and viewing each document as a skip-gram, WeTe can discover more comprehensive embeddings. We compare WeTe with the GloVe by visualizing top-10 words more closer to the same query words at below. Compared to glove, the word embedding we learned tends to be more semantically diverse. For example, "download", "modem" for "pc", and "goal", "win" for "game".  We attribute this to the document level context.
> >
> > | Method(Query word) |                                 Top words                                |
> > |:------------------:|:------------------------------------------------------------------------:|
> > |      GloVe(pc)     |  desktop computer software macintosh computers pentium pcs microsoft xp  |
> > |      WeTe(pc)      |     pc desktop macintosh pcs microsoft internet os download mac modem    |
> > |     GloVe(game)    |      game games season play match player league team scored playoffs     |
> > |     WeTe(game)     |          game season play match team playoff bowl goal win coach         |
> > |    GloVe(world)    |    world cup international olympic european championships event europe   |
> > |     WeTe(world)    |    world cup international european event asian asia women nation team   |
> > |    GloVe(school)   | school college university schools students education elementary graduate |
> > |    WeTe(school)    | school high student campus district church program degree taught harvard |
> >
> > We have added the above discussion in Appendix E in our revised version. Please refer to Appendix E for more detailed description.

---

### Official Review · Reviewer_eWQp · 2021-11-03

**Correctness:** 3
**Technical Novelty And Significance:** 2
**Empirical Novelty And Significance:** 2
**Recommendation:** 5
**Confidence:** 4

**Main Review:**

Strong points:

- While the combination of topic models and word embeddings has received substantial study, it remains an interesting problem.  An optimal transport (or conditional transport) approach such as the one proposed is a promising direction.

- The ability to initialize with pre-trained word embeddings and fine-tune the model is valuable (though this paper is not the first to do this).

- Experimental results versus a range of baseline methods are encouraging.

- The qualitative results on embeddings of words and topics were nice to see.

Weak points:

- My biggest complaint with this paper is that the necessity and motivation for the proposed method is unclear.  While combining topic models and word embeddings is a worthy problem, and transport methods are an interesting potential solution, this paper is far from the first to study the intersection of topic models and word embeddings.  This is already a crowded space which includes the cited papers, such as the ETM and neural topic models, and several others as well.  The present manuscript doesn't make a compelling argument that the proposed method resolves urgent problems in this area.  Furthermore, the proposed method is stated then studied without a principled justification. It reads like the authors wanted to try this method, so they did, rather than that there was a particular reason or mathematical justification for why the method is designed the way that it is.

- The manuscript was quite hard to follow at times, partly because the lack of clearly stated justification for the proposed approach leads to substantial effort to understand what exactly the authors are doing.  The use of non-standard notation in some instances was a contributing factor in this issue.

- Transformer-based language models such as BERT have become the state of the art for representation learning in natural language processing in recent times.  This work should ideally compare with BERT, etc, or justify why it does not do so.  It also needs to discuss the merits of the proposed approach versus transformer-based language models.

- Since the objective function (Equation 13) combines the conditional transport objective with a likelihood objective, both an approach using conditional transport alone and an approach using likelihood optimization alone should be compared to as baselines.

- Separately encoding transport costs c(w, \phi) and distances d(w, \phi), which are "related but also with differences" (pg 4), seems inelegant.

- If speed advantages are claimed (cf. pg 6, "more efficient learning"), this should be demonstrated with rigorous experiments.

- Hyperparameter tuning could have been done more systematically for the experiments, rather than using fixed values for most experiments, followed by a brief sensitivity test.

- The training algorithm could have been explained more clearly, and in the paper itself instead of in the appendix.


Additional feedback / minor suggestions:

Pg 6, Equation 6, it looks like there is an issue in the last step, which doesn't clearly follow from the previous step.  I think there is a missing term on the top and bottom here corresponding to P_j(W_{ji}).  Compare to Equation 9, which has \theta terms on the top and bottom of the last equation in the line.

Pg. 10 (references) - "Latent dirichlet allocation" needs a capital "D."

Pg. 12 (Appendix), "the training algorithm of our WeTe is shown in Algorithm 1" needs a capital T.

A couple of highly relevant missing references:
K. Keya, Y. Papanikolaou, and J. R. Foulds. Neural embedding allocation: Distributed representations of topic models. ArXiv preprint arXiv:1909.04702 [cs.CL], 2019.

J. R. Foulds. Mixed Membership Word Embeddings for Computational Social Science. Proceedings of the 21st International Conference on Artificial Intelligence and Statistics (AISTATS), 2018.

**Summary Of The Paper:**

The manuscript proposes a combined topic modeling and word embedding method.  The idea is to define word and topic embeddings in the same space, represent documents based on both of them, and use a transport-based method to train these two representations to be similar to each other.  The final objective function combines a "conditional transport" objective with a log-likelihood term, and is optimized via Adam.  Experiments find that the method achieves good topic coherence scores and document clustering metrics compared to baselines.


**Summary Of The Review:**

While the proposed method has potential and there are some positive aspects of the work, the manuscript is not ready for publication due to deficiencies in several aspects, including: motivation of the need for the general approach, justification of the specific choices in the approach, other clarity issues, relationship to BERT and transformer-based language models, and certain aspects of the experiments.

---

> ### Author Response · Authors · 2021-11-22
> **Response to Reviewer eWQp (1/2)**
>
> Thank you for your time and detailed feedback. The typos and relevant missing references have been addressed in the revised version, with the major changes  marked in blue.  Below we respond to your questions/criticisms.
>
> First of all, after rechecking Eq.(6), we confirm it is correct. Eq.6 denotes the conditional distribution that specifies how likely a given topic $\mathbf \alpha_k$ will be transported to word $\mathbf w_{ji}$ in document j:
> \begin{equation}
> \pi_{N_j}(\mathbf w_{ji}| \mathbf \alpha_k)=\frac{P_j(\mathbf w_{ji})e^{-d(\mathbf w_{ji},\mathbf \alpha_k)}}{\sum_{i'=1}^{N_j}P_j(\mathbf w_{ji'})e^{-d(\mathbf w_{ji'},\mathbf \alpha_k)}},
> \end{equation}
> where the empirical distribution $P_j(\mathbf w_{ji})$ is defined as the Uniform prior of words in document j: $P_j(\mathbf w_{ji})=1/N_j$ (Eq.3 in our manuscript). By simply deducing, we can obtain Eq.6 in the manuscript:
> \begin{equation}
> \pi_{N_j}(\mathbf w_{ji}| \mathbf \alpha_k) = \frac{e^{-d(\mathbf w_{ji},\mathbf \alpha_k)}}{\sum_{i'=1}^{N_j}e^{-d(\mathbf w_{ji'},\mathbf \alpha_k)}},~~~\mathbf w_{ji}\in\{\mathbf w_{j1},\ldots,\mathbf w_{jN_j}\},
> \end{equation}
>
> Although both Eq. 6 and Eq. 9 represent conditional transport distributions, they represent different roles. Eq. 6 denotes the topic-to-doc transport distribution, and takes the uniform word prior as the weights, while Eq. 9 denotes the doc-to-topic transport distribution, it takes the topic proportion as the weights. The former can be removed from the fraction, while the latter cannot.
>
> **Q1**
>
> Thanks for your suggestions. We have clarified our motivation in the Introduction in the revised version and explain it below.
>
> Rather than just combining topic models and word embeddings, we propose a new view to learn topic model, which is quiet different from previous ones. BPTMs usually are less scalable for big corpora and need to be customized accordingly. Despite the flexibility and scalability in NTMs, they typically need to use approximate posteriors for satisfying the reparameterization trick, potentially introducing additional complexity or approximation errors.
>
> To address the above shortcomings, we propose a novel topic modeling framework in an intuitive and effective manner, which has the same goal with TMs for learning the global topics and the document-specific topic proportions. However, without building an explicit generative process, we formulate the learning of topic model (e.g., optimizing the likelihood) as the process of minimizing the distance between each observed document $j$ and its corresponding trainable distribution. More specifically, the former (document $j$) can be regarded as as an empirical discrete distribution $P_j$, which has an uniform measure over all the words within this document. To construct the latter (trainable distribution), we can represent $P_j$ with $K$ shared topics and its $K$-dimensional document-specific topic proportion, defined as $Q_j$, where we view shared topics as $K$ elements and topic proportion as the probability measure in $Q_j$. It is very reasonable since the $k$-th element in topic proportion measures the weight of topic $k$ for a document, and the document can be well represented using the learned topic proportion and topics from a desired TM. Recalling that each topic and word are usually live in the $V$-dimensional (vocabulary size) space in TMs, it might be difficult to directly optimize the distance between $P_j$ and $Q_j$ over $V$-dimensional space. Motivated by  Dieng et al. (2020),
>
> we further assume that topics and words live in the same embedding space, much smaller than vocabulary space. By abuse of notation, we still use $P_j$ over the word embeddings and $Q_j$ over the topic embeddings as two representations for document $j$. Therefore, we turn towards pushing the document-specific to-be-learned distribution $Q_j$ to be as close as possible to the empirical distribution $P_j$. Although there are several limited works that also view the topic modelling as minimizing distance between distributions, we here view a document as a mixture of word embeddings and a mixture of topic embeddings, which is very different view from these work. Benefiting from our novel view, we can further consider the bi-directional transport between $Q_j$ and $P_j$ across all documents. These fundamental differences lead to different views of topic modelling and different frameworks as well.

---

> > ### Author Response · Authors · 2021-11-22
> > **Response to Reviewer eWQp (2/2)**
> >
> > **Q2**
> >
> > We have modified some notations for easy reading in the revised version.
> > For example, $\mathbf \alpha$ for topic embeddings, and $\mathbf \alpha_k$ for topic distribution.
> >
> > **Q3**
> >
> > We agree on the success of transformer-based language models such as BERT for representation learning. However, although our model can be viewed as a document representation learning approach, ours is different from transformer-based language models. First of all, ours aims to learn topical representations which can be interpreted by the proportion over a set of topics, while BERT like models' learned representations are usually not intepretable.
> >
> > Secondly, these models usually consist of an extremely large set of parameters which need to be trained with large text corpora. While ours is trained on the target corpus with a much smaller architecture (i.e., a few layers of multi-layer perceptions). Therefore, it might not be straightforward to compare ours as a topic model with transformer-based language models.
> >
> > **Q4**
> >
> > In the submitted version, we have performed experiments to explore the parameter sensitivity of WeTe on 20NG dataset with different $\epsilon= [0.1,0.2,...,1]$ in Figure 2. Following your suggestion, we further consider two variants as the degraded versions of our proposed WeTe. The first one only considers the transport cost, called as WeTe(CT); the second one uses the likelihood objective alone, called as WeTe(TM). We have added results of WeTe(CT) and WeTe(TM) on 20NG dataset into Figure 2 in the revised version. We can see that
> > the transport cost leads to high topic quality, and the likelihood has benefits for the representation of documents. By combining these two objectives together, WeTe can produce better performance than using only either of them.
> > We will add more results on all datasets in our final version.
> >
> > |  Method  | Purity |  NMI |  TQ  |
> > |:--------:|:------:|:----:|:----:|
> > | WeTe(CT) |  60.2  | 30.0 | 6.20 |
> > | WeTe(TM) |  63.5  | 31.4 | 1.32 |
> > |   WeTe   |  **67.3**  | **35.0** | **6.24** |
> >
> > **Q5**
> >
> > Thanks for your suggestion. Since we have explained the  $d(\mathbf w_{ji},\phi_k)$ in Eq (6) and $c( \mathbf w_{ji},\phi_k)$ in Eq (7), we  removed this sentence in our revised version.
> >
> > **Q6**
> >
> > Following your suggestion, in the revised version, we have conducted several experiments on the large scale RCV2 dataset to validate the efficiency of our proposed model and reported these results in Figure F1 in Appendix F. We found that the proposed model has about the same convergence speed as other neural topic models. We have revised those similar statements in our revised version.
> >
> > **Q7**
> >
> > We need to clarify that our model has very limited hyper-parameters, including trade-off hyper-parameter $\epsilon$, learning rate batch-size, number of topics. In this paper, we aim to demonstrate the effectiveness of our proposed WeTe in terms of learning topics and document representation rather than exhaustively tuning those hyper-parameters. Therefore, we choose routine settings for these  frequently-used hyper-parameters except for the  trade-off hyper-parameter $\epsilon$. Notably, for hyper-parameter $\epsilon$, we had explored the  sensitivity of our proposed model at Fig. 2, and observed that our proposed model is robust to the choice of $\epsilon$, especially for the topic quality. Besides, one can achieve even better performance by fine-tuning $\epsilon$ for each dataset carefully, which however is not our main point.
> >
> > **Q8**
> >
> > We will move the algorithm into the main body in the final version.

---

> > > ### Comment · Reviewer_eWQp · 2021-11-30
> > > **Thanks for the response**
> > >
> > > Thank you to the authors for the extremely detailed response, and for the updates to the paper.  I will take this on board.

---

> > > > ### Author Response · Authors · 2021-12-02
> > > > **Re: Thanks for the response**
> > > >
> > > > Thank you for your kind reply. Please let us know if you have any additional comments or suggestions.

---

### Author Response · Authors · 2021-12-02
**Feedback from reviewers**

Dear Reviewers,

We appreciate it if you could let us know whether our responses and revisions are able to address your concerns.

Thank you,

Paper786 Authors

---

### Decision · Program_Chairs · 2022-01-20

**Decision:**

Accept (Poster)

**Comment:**

This is a borderline paper on the well researched theme of Topic models.
The strongest point of the paper is that it proposes a new topic modelling framework where both word and topic embeddings live in the same space.
 It then appeals to optimal transport theory to do the necessary training using SGD. However, this is not the first paper to examine
Topic models and Optimal Transport theory. Several papers[1,2,3] in the recent past have started investigating this line of research.  In the rebuttal phase, the author(s) justify the choice of state of the art methodologies and also discuss the key conceptual difference between
existing literature and the submitted one. The major difference seems to
they approach the problem differently leading to better quality topics(as measured by several metrics) and computational efficiency---existing state of the art requires more complicated iterations whereas proposed approach works with SGD.
The manuscript, if accepted, needs to be updated considerably to reflect some of these aspects.




[1] Hongteng Xu, Wenlin Wang, Wei Liu, and Lawrence Carin. Distilled wasserstein learning for wordembedding and topic modeling. In NeurIPS2018.

[2] Viet Huynh, He Zhao, and Dinh Phung. OTLDA: A geometry-aware optimal transport approach fortopic modeling. In NeurIPS2020.


[3] He Zhao, D Phung, V Huynh, T Le, and W Buntine. Neural topic model via optimal transport, 2020